# Developing Institutions and Inter-Organizational Synergies through Digitalization and Youth Engagement in African Agriculture: The Case of "Africa Goes Digital"

Matthew Ayamga [1,*], Abdelaziz Lawani [2,*], Selorm Akaba [3,*] and Arsene Birindwa [4]

1　Business Management and Organization (BMO) Chair Group and Information Technology (INF) Chair Group, Wageningen University & Research, P.O. Box 8130, 6700 EW Wageningen, The Netherlands
2　Agribusiness Management & Entrepreneurship, Department of Agricultural and Environmental Sciences of Tennessee State University, 202K Farell-Westbrook Complex, 3108 John A. Merrit Blvd, Nashville, TN 37209, USA
3　Department of Agricultural Economics and Extension, School of Agriculture, College of Agriculture and Natural Sciences, University of Cape Coast, Cape Coast CC-075-8216, Ghana
4　Van Hall Larenstein University of Applied Sciences, Department of Development Studies, Larensteinselaan 26-A, 6882 CT Velp, The Netherlands
*　Correspondence: matthew.ayamga@wur.nl (M.A.); alawani@tnstate.edu (A.L.); sakaba@ucc.edu.gh (S.A.)

**Abstract:** As digitalization increasingly gains attention in the agriculture sector, many African youths are seizing the opportunity provided by digital technologies to engage in agriculture. While prior studies have started examining the intricacies of digitalization in agriculture, they have paid less attention to its implications for fostering institutions and inter-organizational synergies that can support and sustain the businesses of youth enterprises engaging with Africa's agriculture. Against this backdrop, we undertook a qualitative case study to shed light on the development of a community of practice comprising youth enterprises engaging in agriculture through digitalization. Thus, we studied Africa Goes Digital (AFGD), a consortium of young African enterprises that rely on digital technologies and solutions to provide services to farmers across the African continent. The results showed that: (1) the Technical Centre for Agricultural and Rural Cooperation ACP-EU (CTA) played a pioneering role as a development agency in supporting enterprises with drones and training for agriculture, which led to the formation of AFGD; (2) two prevalent themes characterized the services offered by AFGD members; (3) AFGD brings members under one platform where they leverage WhatsApp for communication and knowledge sharing and Twitter to showcase success stories and form partnerships with each other to execute certain development projects with governments and development agencies; (4) the lack of regulations or the restrictive regulations on the use of drones are still problems that some members face, and the high costs of sensors and related drones are challenges for the members. Building on the results, we discuss the implications of digitalization for supporting African governments in meeting sustainability goals and conclude by describing the theoretical contribution of our study and promising future research directions.

**Keywords:** digital innovations; sustainable development; digital agriculture; self-employment; topic modelling; entrepreneurial ecosystems; community of practice; CTA

## 1. Introduction

It has been suggested that the Fourth Industrial Revolution and digitalization will enable Africa to transform into a global powerhouse [1]. In particular, digitalization will have a role in the transformation of African agriculture [2]. Extant studies and institutional reports on digital technologies in agriculture have highlighted the potential of digitalization to drive the transition toward more sustainable agriculture [3–6][1].

As Africa's economic backbone is agriculture, with about 70% of its population depending on agriculture for their livelihoods [7], agricultural development in Africa in recent

years has been transitioning from traditional agricultural practices and marketing norms to more efficient, sustainable, inclusive, transparent and resilient agriculture based on digitalization [8,9]. Digitalization in agricultural development initiatives, such as those concerning agriculture and food security in rural landscapes, poverty alleviation, youths in agriculture, climate change adaptation, ecosystem conservation, etc. [10], has often been framed as disruptive or transformative [11–13]. This is because the use of digital technologies in agriculture can create new forms of value and new types of market while, on the flip side, changing the way current agri-food production systems function [14–17].

Governments and development agencies have also touted digitalization in Africa's agriculture as a way to address unemployment, ensure food security, provide market linkages and reach dispersed smallholder farming communities with agricultural advisory services [18]. Indeed, digitalization in agriculture in Africa is seen as a game changer. It attracts educated youth who rely on digitally driven innovations to provide services to farmers, enabling them to overcome emerging climate change challenges [16].

Scholars have started examining the implication of digitalization in Africa's agriculture. Recent studies have focused on the need for regulatory frameworks [19,20], farmers' perceptions of digital technologies [21] and the socio-technical factors influencing the adoption of digital technologies [22,23]. Prior studies also have examined the entrepreneurial activities surrounding the adoption of digital innovations, mostly focusing on why businesses fail [24]. Though digitalization in agriculture promotes entrepreneurial activities, successfully implementing digital innovations in the long term is a challenge in most African countries. Madichie et al. [25] attribute this challenge to the lack of digital infrastructure—poor internet access and lack of electricity in remote farming communities—which prevents the successful deployment of digital innovations.

Nevertheless, the transformative potential of digitalization in agriculture in fostering self-employment practices, particularly for educated youths, and the drive to succeed by learning from each other has received less attention. It has been observed that digital innovation success in Africa's agriculture must be organized along two pathways. First, it must be well-defined as part of wider agricultural development initiatives [26]. Second, the digital transformation needs to occur within a wider agritech ecosystem including different actors—farmers, digital technology service providers, governments and development agencies [25]. Indeed, in this context, it has been suggested that agritechs must begin to form communities of practices (COPs) to enable them to (1) attract the right government support and favourable policies [27], (2) secure investments to scale up their services [28], (3) learn new ways of innovating [29] (establish the right business models that align the services provided with the needs of farmers and other actors within the agriculture sector) and (4) share knowledge and skills relating to the innovation process among innovation agents [30]. COPs enable collaborative activities, which have become the core of inter-organizational relationships [31]. Institutional arrangements govern the collaborations in such COPs and are enacted by the members to secure the collective benefits of the communities [32].

The emergence of new COPs surrounding digitalization in agricultural development in Africa implies that individual enterprises leveraging digital technologies recognize the need for learning through the digital innovation process [33]. Sharing experiences in the digital innovation process is a way of learning, and the formation of a COP serves as the means for such experiences to be shared [34]. However, the organization of a COP is not an easy task, especially when individual members are constrained in making their contributions to achieving the common goal of the COP.

In response to these calls, this study aimed to understand the development of COPs leveraging digital innovations for agriculture on the African continent and to ascertain how they promote institutions and inter-organizational synergies, as well as the related shortcomings. In achieving these objectives, we employed Africa Goes Digital[2] (AFGD) as our case study. Accordingly, we aimed to answer the following research questions (RQs):

RQ1: How was AFGD established and what are the prevalent themes characterizing the services offered by its members that leverage digital innovations?

RQ2: How is AFGD developing institutions and inter-organizational synergies to help its members enhance their digitally driven services for farmers?

RQ3: What are the challenges and opportunities for both new and old enterprises in the digital revolution in Africa's agriculture?

In the remaining sections of this paper, we present the entrepreneurial ecosystem surrounding digitalization in Africa (the emergence of a community of practice), followed by our methodological approach and the results. Then, we discuss the findings, the paper's theoretical contributions and future research directives. Finally, we present the conclusion.

## 2. Entrepreneurial Ecosystems—The Emergence of a Community of Practice in Africa through Digitalization

The ecosystem perspective has been recommended for studying entrepreneurial activities surrounding digital enterprises [25,35,36]. Scholars have argued that there are three forms of entrepreneurial ecosystems related to digital evolution [37]—business ecosystems, innovation ecosystems and platform ecosystems. These ecosystems all hold in common interaction between parties to secure and harness more productive entrepreneurial outcomes [38–40]. They focus on the interactions between dynamic actors within a community of enterprises creating value for targeted customers [41,42].

Entrepreneurship is a crucial solution to some African countries' socio-economic and environmental challenges [43]. Accordingly, entrepreneurial ecosystems are regional communities of enterprises that can provide insights into the exchange of knowledge and business model innovations [44]. However, research on entrepreneurial ecosystems in Africa is in its infancy, particularly with regard to the emergence of digital enterprises. Most entrepreneurship-related studies in Africa have focused on describing the state of entrepreneurship on the continent. They neglect to capture the interactions and coexistence between enterprises, policymaking and the entrepreneurial environment [25]. Sheriff and Muffatto [45] point out that studying entrepreneurial ecosystems can provide a means for enacting policies for entrepreneurship in Africa.

With the digital revolution in Africa's agriculture, huge amounts of data have been generated annually based on information and data exchange and interlinkages across the agricultural space, which has resulted in the emergence of a COP [46]. As this is a COP based on the use of digital technologies in agriculture, headway can be made in transitioning to a more sustainable agriculture [47]; that is, as a community. The interplay between the enterprises involved in the digital evolution can help us to understand how they foster knowledge sharing [44], enabling the constant updating of the strong knowledge base required for sustainable development [48], providing new ways of conceptualizing and measuring impact [47] and drawing governments' interventions and the needed support from development agencies. In what follows, we analyse AFGD's reliance on digital innovations in Africa in order to provide cutting-edge services to farmers. By so doing, we show what makes a successful entrepreneurial ecosystem of enterprises in African agriculture influenced by digitalization.

## 3. Materials and Methods

### 3.1. Research Design

We employed a qualitative case-based research approach in this study. We used a case study design with a single case—AFGD, a community of enterprises offering digitally driven services to farmers across Africa. As defined by Stake [49] (p. 27), a single-case study is *"an intensive, holistic description and analysis of a single instance, phenomenon, or social unit"*. A case study also allows for an in-depth study over time of the development of a phenomenon in a specific geographical area [50].

### 3.2. Data Collection and Analysis

The data collection for this study was undertaken in two parts. The case study design allowed us to use multiple approaches for data sourcing and triangulation and generate strong evidence [51]. First, we requested interviews with some of the identified members of AFGD who had been evaluated and confirmed to still be members of AFGD, and only five of them responded (see Section 5.2 for the limitations of our study). Then, we held in-depth interviews with the five representatives of AFGD who served as our key informants to understand the development of AFGD and shed light on its role in promoting institutions and inter-organizational synergies in the digitalization of Africa's agriculture. We completed the interviews using a semi-structured interview guide (see Appendix A). Furthermore, notes were taken during the interviews, which lasted between 36 min and 1.2 h. This further enriched our database, as the interviews were recorded and transcribed.

Second, we complemented the interviews with secondary data sources—a desk analysis of the activities of AFGD and its members via their websites and the related posts on their social media pages (e.g., Twitter). When analysing the data, we employed inductive coding through active reading, highlighting of important phrases in the transcripts and the application of open codes, which resulted in the identification of several relevant themes, as shown in the results section and supported by quotations. In addition to the qualitative analysis, we conducted topic modelling to enable us to answer RQ1.

Topic modelling is a statistical tool popular in natural language processing that is used to understand collections of texts [52]. This unsupervised learning technique employs the correlations between words to group them into topics. For this study, we used the latent Dirichlet allocation (LDA) method developed by Blei et al. [53]. LDA is a generative probabilistic model that identifies latent topics in a collection of texts. Here, we represented the description of services by each member of the association as a vector of word counts (a bag of words). Each description had a probability distribution around several topics and each topic had a probability distribution around several words. The LDA requires the researcher to set the number of topics. There is no agreement on the measure of the right number of topics [54]. In our study, we iterated through several numbers and retained two (02) topics, as the texts were short and this number produced more coherent models.

The outcomes of our topic modelling are shown in Section 4.1 being generated using Google Colaboratory (Google Colab). Bisong [55] has examined the use of Google Colab for prototyping machine learning models.

We present a summary of this single-case study and the related methods employed to answer our research questions in Table 1.

**Table 1.** Case study design.

| Case Study Design Activities | Explanation |
|---|---|
| Goal | To understand the development of a COP (AFGD) leveraging digital innovations for agriculture on the African continent and to ascertain how that promotes the growth of institutions and inter-organizational synergies, as well as the related shortcomings |
| Research Questions (RQs) | RQ1: How was AFGD established and what are the prevalent themes characterizing the services offered by its members that leverage digital innovations? <br> RQ2: How is AFGD developing institutions and inter-organizational synergies to help its members enhance their digitally driven services for farmers? <br> RQ3: What are the challenges and opportunities for both new and old enterprises in the digital evolution in Africa's agriculture? |
| Background and Source | Website of Africa Goes Digital <br> Key informant interviews <br> Documents and related reports <br> Observation |
| Data Collection | Direct data collection using interviews and indirect data collection using documents and reports analysis |
| Data Analysis | Qualitative and topic modelling analyses |

## 4. Results

In the following sections, we present the answers to each of the RQs formulated and focus on an analysis of how AFGD was established and the prevalent themes characterizing the services offered by its members that leverage digital innovations (RQ1), how AFGD develops institutions and inter-organizational synergies to help its members enhance their digitally driven services for farmers (RQ2), and the challenges and opportunities for both new and old enterprises in the digital evolution in Africa's agriculture (RQ3).

### 4.1. RQ1: AFGD and the Prevalent Themes Characterizing the Services Offered by Its Members

Fundamentally, AFGD came into existence following the pioneering efforts of the Technical Centre for Agricultural and Rural Cooperation ACP-EU (CTA). The CTA supported and launched drone operations in about 11 African countries. The enterprises supported in the 11 African countries came together to share their experiences in an experience capitalization workshop organized by the CTA in 2018 in Ghana. It was after the experience capitalization workshop that the idea of AFGD was conceived. The experiences of the 11 enterprises ranged from leveraging mobile phone applications to remote sensing of images captured by satellites and/or drones. These experiences also extended to applying drone technology and other digital solutions to support the conservation of natural forests, limiting the effects of farming activities. AFGD was officially incorporated in October 2019 in Delaware, USA, as a non-profit corporation. One member stated:

> *"since it's a technology (drone technology) that is new and evolving, it's good to keep together and learn from each other. There are kind of different cases that we all apply within our different countries"*. (AFGD member from Tanzania)

AFGD started with 11 enterprises from 11 African countries in 2019 and has grown to about 44 members from 22 different African countries in 2022. However, in this study, we validated only 35 members. These members were all young, educated individuals leveraging digital technologies—especially drone technology—as the foundation of their enterprises and aiming to change the narrative about youth engagement in Africa's agriculture.

> *".. the members are actually African companies that are run by the youth. So here these African companies leverage the power of digital technologies such as Unmanned Aerial Systems (UASs), satellite imagery and GIS to deliver quality services and high-end products that will eventually help solve problems for African farmers and other clients that will benefit from these digital services.."* (AFGD board member from Kenya)

We present the prevalent themes characterizing the services[3] offered by AFGD members as a word cloud in Figure 1. It provides an overview of the words used in the descriptions. The words that occur more often are represented with a font size that corresponds to their frequency.

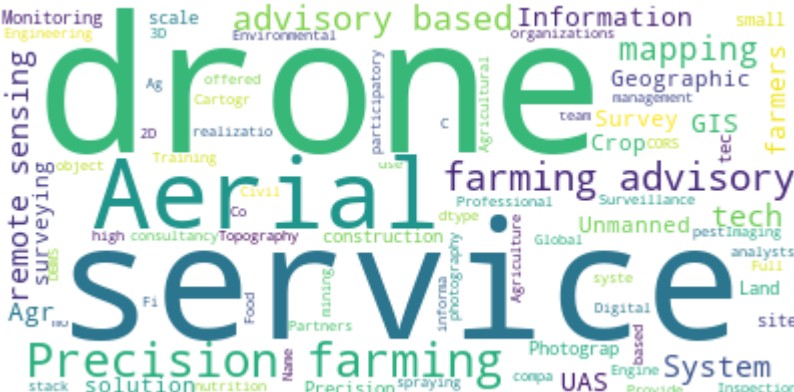

**Figure 1.** Word cloud of the descriptions of services by members of Africa Goes Digital.

Table 2 presents the two topics identified in our study and the keywords that defined them based on their probability of occurrence in the topics.

**Table 2.** Topics and corresponding keywords.

| Topic Number | Topic Percentage Contribution | Representative Keywords |
|---|---|---|
| 1 | 0.9914 | Drone, survey, mapping, service, precision, training, base, aerial, GIS, agriculture |
| 2 | 0.9927 | Remote, management, decision, precision, monitor, information, monitoring, sensing, support, plot |

A weakness of topic modelling using LDA is the inability of the algorithm to label the topics. Using representative keywords, researchers can infer the broad categories for each topic. This inference can be challenging, since some words can appear in several topics. To better understand fitted LDA models, Sievert and Shirley [56] developed a web-based interactive visualization tool called LDAvis. LDAvis presents the topics, their differences, and a careful examination of the terms most highly associated with each of them.

In Figure 2, we present an interpretative visualization of our models. The left panel shows the prevalence of each topic (areas of the circle) and how they are related to each other (the distance between them).

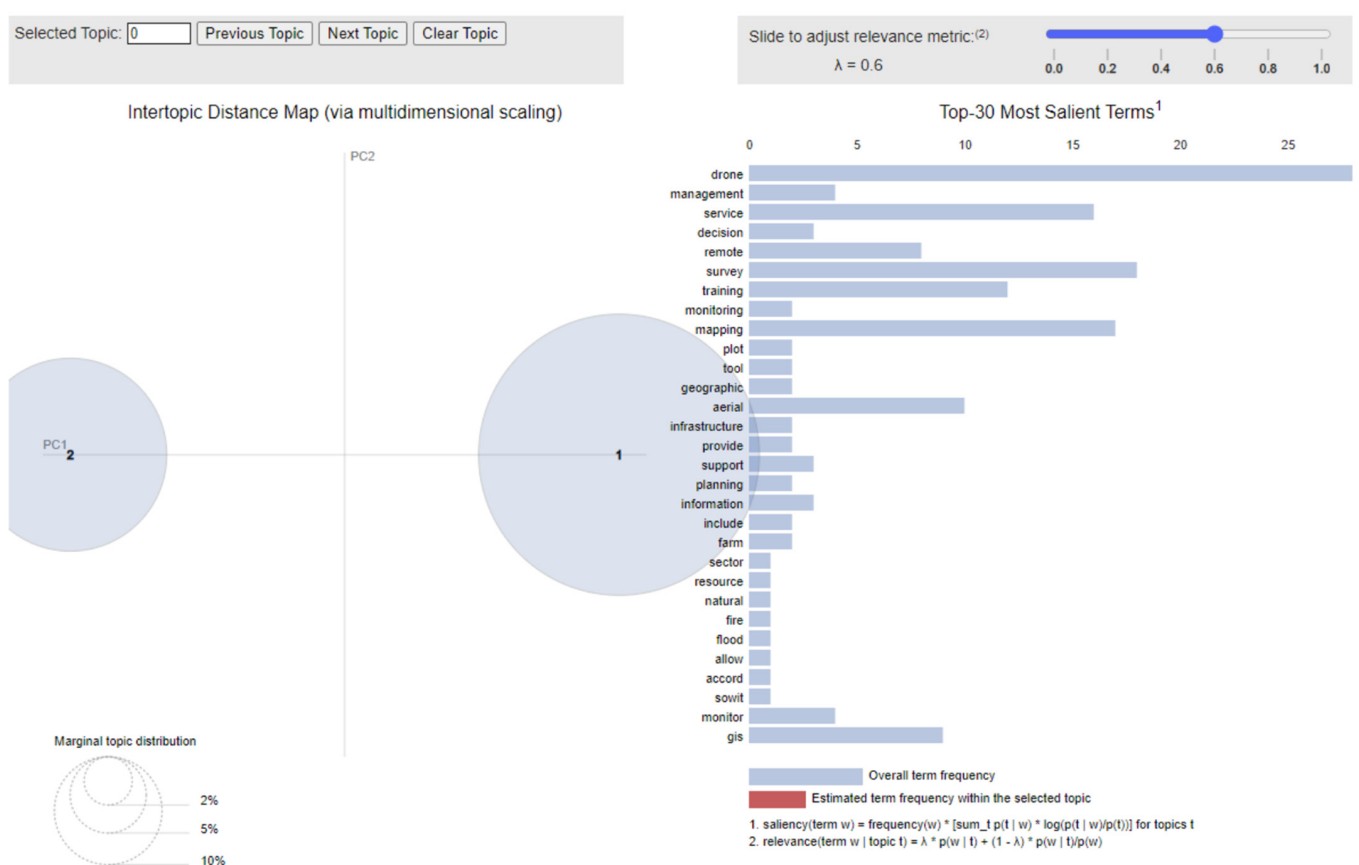

**Figure 2.** LADvis representation of the topics and term bar charts. For the salient terms see Chuang et al. (2012) and for the relevant terms see Sievert & Shirley (2014).

The right panel shows the most common terms. The widths of the grey bars represent the corpus-wide frequencies of each term. When a topic is selected, the right panel shows the terms that are more useful in interpreting that topic. In other words, visualization helps researchers infer the meaning of a topic and the importance of the terms related to the topic.

Figure 3 shows the most relevant terms in topic one and Figure 4 the most relevant in topic two.

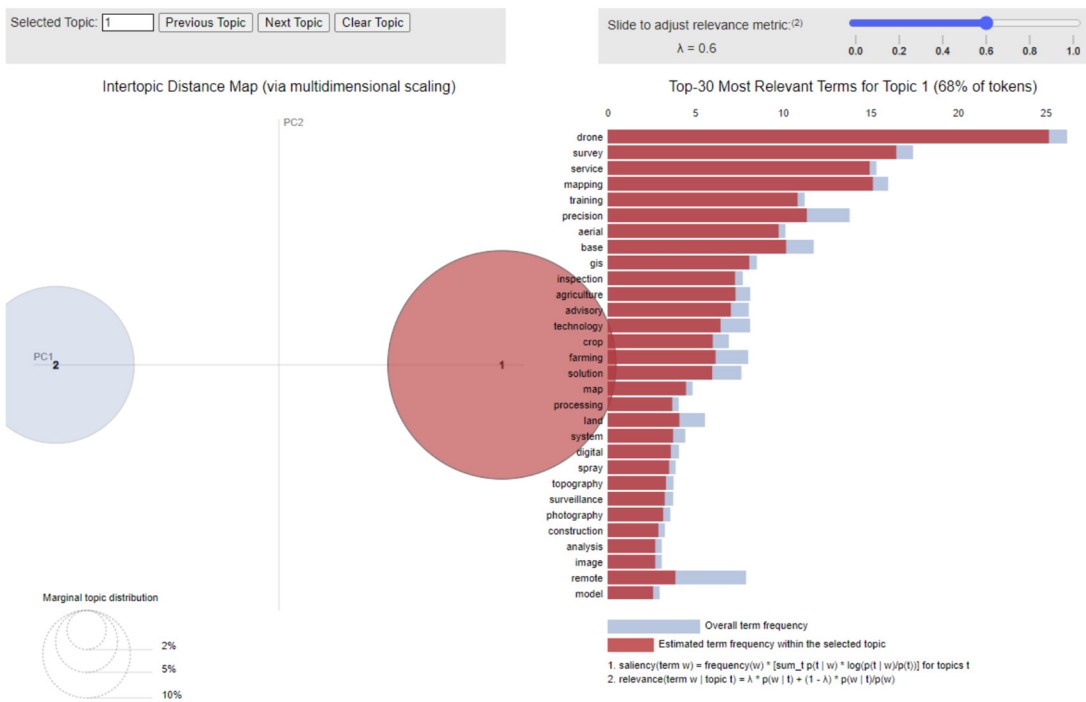

**Figure 3.** Term relevance in topic one. For the salient terms see Chuang et al. (2012) and for the relevant terms see Sievert & Shirley (2014).

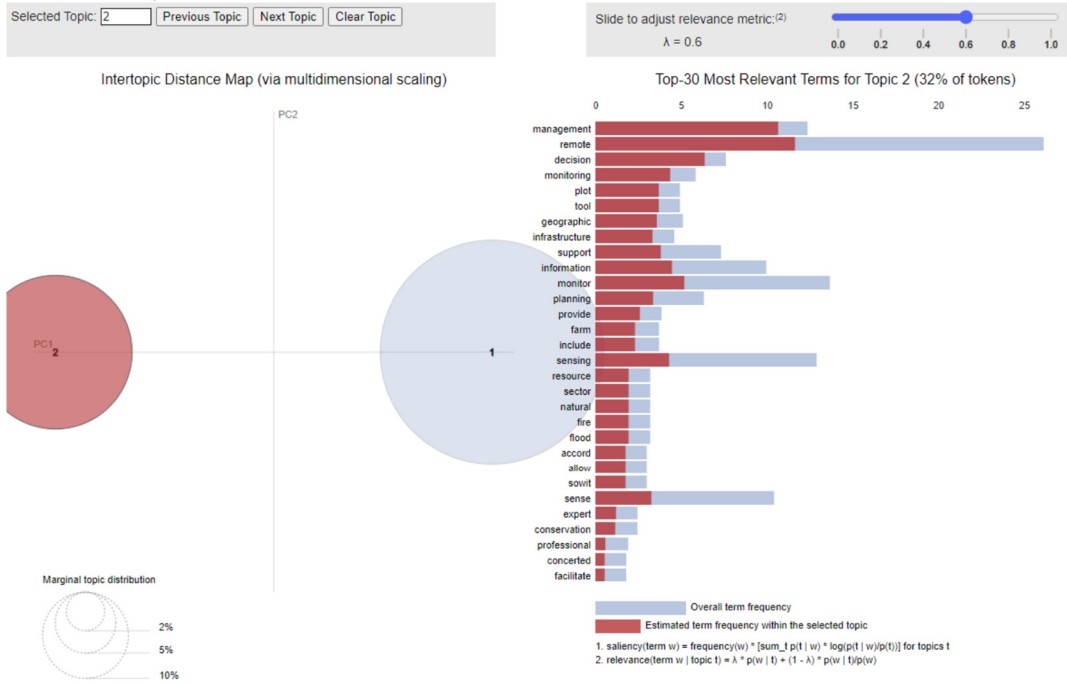

**Figure 4.** Term relevance in topic two. For the salient terms see Chuang et al. (2012) and for the relevant terms see Sievert & Shirley (2014).

As shown in Figure 3, the six most representative terms for topic one were "drone", "survey", "mapping", "service", "precision" and "training". This suggests that topic one is related to the services offered by the members in the area of aerial surveys using drones and training.

As shown in Figure 4, the six most representative terms for topic two were "remote", "management", "decision", "precision", "monitor" and "information", suggesting that topic two is related to services in the areas of management and information provided to farmers as decision-support tools.

Another advantage of the LDAvis tool is that it offers the ability to examine how a term is relevant to a selected topic. For example, the term "remote" (Figure 4) was one of the most important terms characterizing topic two (width of the red bar) but also one of the most frequent terms in the overall description of services (grey bar).

Analysis of the two topics confirms that members of AFGD are involved in providing essentially digital services to African farmers using precision agriculture tools such as drones. The information derived from those services constitutes important decision-support tools for better management of farm operations. These services have the potential to change farming practices in Africa moving it from traditional farming systems where decisions are based on experience and intuition to ones where decision-making is based on evidence.

### 4.2. RQ2: How AFGD Is Developing Institutions and Inter-Organizational Synergies for Its Members

AFGD came into existence through an experience capitalization workshop and, after its formation, the members established a mechanism of collaboration and learning, whereby members who did not have digital–technical expertise relied on members who did have such expertise through partnerships. These partnerships have two main aims. First, they aim to enable the members without digital–technical expertise to meet the requirement of operating their businesses successfully. The requirement of operating successfully relates to the form of business model innovation and capacity building in non-digital–technical enterprises. This kind of inter-organizational relationship between members is controlled by the institutional norms that govern AFGD and prevent exclusion and business overshadowing among its members. It also establishes a sense of inclusiveness and belongingness, where members feel they are part of an ecosystem that collectively finds solutions to the challenges encountered. Secondly, the partnerships enable the members to join forces in securing funding for projects from development agencies that work with rural farmers to improve their livelihoods. Members in partnerships then work together to obtain the project deliverables for the development agencies whilst establishing relationships with farmers with their services. They also enable them to collaborate and work with governments in many government-related projects across various sectors in their respective countries by securing and supporting the interventions needed for sustainable development.

> *"In the respective countries of the members, there are lots of projects going on. They are helping a lot of government interventions in terms of infrastructure, agriculture, health, mining, and surveying. So they are playing key roles. We can look at cases of Charis from Rwanda, and SOWIT in Morocco, you can look at the case of Integrated Aerial Solutions from Nigeria who is a new member who has joined. So they are playing their roles to be able to induce Africans to embrace these digital technologies and to assist in the development sector. Because now, the world is going digital so all these things they are doing are contributing to rising the application of digital technologies in Africa".* (AFGD technical officer from Ghana)

> *".. members are playing different roles in their respective countries in that every country is faced with a unique type of challenge. So, we have one member who is based in Nigeria and uses satellite images to be able to offer scouting services to the farmers who are applying WhatsApp subscribing to his services. So, what happens here is that, what he requires from you is that the boundaries of your farm and the crop that you are planting. So, what he will do is that, with the satellite images he is able to monitor the progress or the development of the crops, and with special algorithms, he is able to tell you that this part of your farm is not good—just go and check on this exact spot and see what is happening. So, this becomes helpful if you have a large farm and you are using traditional method of scouting, this will save lots of time..".* (AFGD board member from Kenya)

Learning can be accomplished in many forms through various means. It may be through regular quarterly or annual meetings to update and share experiences. Members of AFGD, however, learn through communication and sharing of opportunities by leveraging social media applications. For instance, members are in a WhatsApp group where they ask questions about how to resolve problems they encounter and get responses from all members who have managed and resolved such problems. As rightly stated by one of our key informants:

*".. now the best medium they are able to learn from each other is the WhatsApp platform. So if someone has a work to do or a project and it is not based in that particular country, the person will put out in the platform 'are there any operators in this country?' and if there are operators there, they will take the conversation from the back door or is there anybody who can provide information on this? And if there is anybody they interact with on these. We have realized members have requested to have meetings one-on-one so experiences can be shared. So is another effective way in which the unity and collaborations can be formed".* (AFGD technical officer from Ghana)

Another mechanism through which the actors share information and success stories is Twitter, with AFGD then taking these stories and reposting them on its website so all members have access. Sharing information and learning about solutions from their peers in the WhatsApp group further equips the members in their collaborations with governments and development agencies in their respective countries. According to our key informant, AFGD is further looking to organize mini-workshops and stakeholder or forum meetings for its members in the coming months or years to further enhance their collaboration and learning from each other.

*" What we do is we bring members together, to be able to have that interaction, share information, share competencies, and share resources for each other to be able to undertake certain types of activities. So on our WhatsApp page, in case some member in a particular country needs some information, the person is able to make the request on the page and people are able to give him feedback. So we are bringing the digital entrepreneurs together on one platform to be able to exchange. Another thing we are also doing is that we also expose them to some opportunities or some investors who are interested in whatever they are doing. So frequent sharing of grants, conferences and awards schemes for which some members are able to apply..".* (AFGD technical officer from Ghana)

AFGD plays a mediating role in securing key accessories and equipment for its members, who rely on these digital technologies and solutions and introduce them to rural farmers, changing the face of agriculture and other development projects in Africa. As part of the mediating role, our key informant indicated that:

*"... and again what we are currently doing now too is getting access to some accessories and equipment which is usually for some of them to get. For instance, the drone batteries, the drone, the remote controllers, and all those hardware that are so difficult to get— we are assisting members to be able to get and equally deliver them to their doorsteps. The equipment is actually some of the things which are scares and very difficult to get and a lot of requests have come in terms of maintenance of some of the equipment. So AFGD is trying to have or have the networks, identifying the sources of who can provide these services to these entrepreneurs at an affordable rate..".* (AFGD board member from Kenya)

As a consortium of different enterprises, AFGD also serves as a matchmaker for its members and potential investors or clients who might be interested in what the members are immersed in with their digital technologies and solutions. In order to function as such, AFGD takes stock of what each member does and constantly updates its websites with these new pieces of information to allow potential clients or investors to see them.

*".. AFGD also collects data on members about what they do, their competencies and posts them on their website so people can be able to access and make a request for the*

*digital or any client who needs the services of the members..".* (AFGD technical officer from Ghana)

A support mechanism that AFGD wants to provide is the award initiative for its members in which they will award USD 1000 to support the winners in further improving their services, websites and other visible activities and storytelling or publications. As confirmed by one of our key informants:

*".. we have this in mind to be able to push all these small enterprises who are not known. So we run this small award scheme in which members have applied and we are at the stage of screening the applications and we will award those who deserve help".* (AFGD technical officer from Ghana)

*4.3. RQ3: Challenges and Opportunities for Both New and Old Enterprises in the Digital Revolution in Africa's Agriculture*

The first challenge mentioned was the lack of enabling regulations allowing the safe usage of various technologies, such as drones, and the procurement and transport of these digital technologies. Though some countries now have regulations in place, there are many shortcomings in the application of various digital innovations.

*"Majority of the countries are sceptical about the usage of these drones but some countries like Ghana, Ivory Coast, Zambia, Tanzania, and Rwanda, through their Civil Aviation Authorities (CAAs) are able to set regulations able to control the use of drones. And those policies are able to serve our members to operate safely and freely. For countries like Ethiopia which we know, they have an issue, and Kenya but gradually that's why AFGD is coming in to meet up with government institutions so there can be a discussion on how digital technologies can favour our digital entrepreneurs"..* (AFGD technical officer from Ghana)

*".. one of the main challenges we are facing is the drone regulations in Africa. So every country is governed by the respective Civil Aviation Authorities (CAAs). So, we are facing a very slow rate with the countries coming up with drone regulations within the African continent. So, this is improving, it is not as bad as it used to be when the idea was incepted but right now, more and more countries are coming up with regulations to enable us to use drones within the African countries' airspace".* (AFGD board member from Kenya)

Deploying these digital technologies successfully is another hurdle that exists in the evolution of digitalization in Africa's agriculture. The adoption rates are not encouraging, indicating that serious sensitization is required to improve them. As pointed out by one of our key informants:

*".. the challenges you know are from the demand and supply aspects, so in terms of demand on the farmers' side, several people have come to meet farmers, promise them the heaven and earth you know, took their money and left. So, you know, farmers are more reluctant these days to get on board of anything. So you have to provide the service free of charge first. Naturally, they want to see the impact, you know, so that means you have to stay to the end, to the harvesting season before you, you know, anybody will be able to buy into whatever you have".* (AFGD member from Nigeria)

*".. another challenge we are facing is the reception we are getting from small-scale farmers. Because the small-scale farmer has a very tide budget, and the small-scale farmer is used to doing things in a certain way and then you will find most of them have only one planting season because they can't afford to have two planting seasons or maybe the weather is not conducive—so they are like on a time bomb. So, if you do something different, they are afraid things won't go the right way—so, they are not willing to experiment with the use of technology so the update of our services by small-scale farmers is a little bit lagging behind. But we have a mitigation to this. We deal with cooperatives, so cooperatives help them to cushion against some of these risks".* (AFGD board member from Kenya)

The costs for some drones and sensors are also high and, as an enterprise, one needs to have at least two or as mentioned by a respondent: "*the agriculture sensors, they are very costly*" (AFGD member from Tanzania).

However, there are huge opportunities that exist for new enterprises joining the digitalization agenda in agriculture. Given the awareness of the developments in the digital agriculture space, significant demand for digital services will soon arise that the current enterprises cannot meet. Furthermore, the costs of some digital technologies are decreasing, which new enterprises can take advantage of.

> "*.. for example, we were using drones, the price of drones has reduced drastically and you can get a drone depending on the budget that you have. So, the cost of acquiring equipment has gone down and then we also have a lot of online courses that are able to expand your knowledge on how this digitalization can work, some of them are easily accessible and free and we are able to share among ourselves*". (AFGD board member from Kenya)

What is also expected from AFGD is that:

> "*AFGD should also strengthen their partnership with service providers so they can make software available to members. This is in their charter but it's not actioned yet. With yearly subscriptions from members, it allows members to have access to this software or services so that the businesses in the consortium can experience accelerated growth, while the service providers too are promoted across Africa*". (AFGD member from Nigeria)

## 5. Discussion

The digitalization agenda in the agriculture sector demands that attention be divided between: (1) the focal actors who are introducing digital technologies and solutions to the rural farmers; (2) the governments and developing agencies that seek to release the full potential of digitalization in agriculture for sustainable development; and (3) the benefiting farmers, who are at the core of every discourse on sustainable development in Africa. There are some serious implications we can draw from our findings that could provoke a shift in how we view matters related to creating jobs in agriculture; orchestrating engagements in providing the enabling environment for sustainable development across three dimensions (economic, social and environmental); and, finally, the role of the private sector (i.e., looking at entrepreneurship in digital agriculture) and communities of practices, focusing on consortiums aimed at pushing the digitalization agenda in Africa's agriculture.

First, our case study of AFGD showcases the transformative potential of digitalization in attracting new actors to engage with the agriculture sector. Our analysis revealed that the enterprises relying on digital technologies and solutions to provide services to farmers across Africa are young. These enterprises are established and managed by young, educated individuals who are motivated and fascinated by digital technologies that can revolutionize the agriculture sector and create growth opportunities for them and the farmers. The status quo thinking that "farming is engaged by the aged and uneducated who rely on manpower tools" has been changing in recent years. The narrative is now that various young enterprises are leveraging digital technologies to provide services, such as input advisory and market linkage services, to farmers to enable them to address pest issues and increase yields and then link them to already existing markets for their farm products. While agricultural advisory and market linkage services may appear as the predominant use cases, other enterprises are focusing their solutions on curbing post-harvest losses and on food transportation. Digital technologies not only provide job opportunities for these enterprises and entrepreneurial development but also enable them to employ other young individuals in their enterprises, which has helped some governments to achieve some Sustainable Development Goal (SGD) targets.

Second, given the entrepreneurial opportunities offered by digitalization via digital technologies, the findings of this study also set the space for discussions regarding the current global environmental change, which poses huge risks to agri-food systems and the

livelihood of individuals across the globe, and how these new actors (enterprises offering digital services) can help agri-food systems address the challenges. On the flip side, the services offered by the enterprises to farmers are not free and digital technologies are no less expensive. This point implies three things: digitalization must be promoted across agricultural value chains, digitalization must be leveraged to transform the agri-food sector in Africa, and all actors involved in the digital revolution in agriculture must begin to collaborate in order to balance and cope with the dynamics of the ecosystem following the introduction of digital technologies and solutions [57]. By addressing these implications, governments can play their part in providing the needed support, establishing enabling business environments for these enterprises and playing the role of orchestrators in the adoption of digitalization for sustainable development. Moreover, these enterprises do not want to fail by going out of business and will, therefore, adopt new ways of doing business (new business models) and new partnerships, since "business-as-usual" will not work in the current socio-economic context. From our findings, we can suggest that these enterprises could learn to adapt their business models through information sharing in the community of practice and by employing collaborative mechanisms with other actors to stay viable. Such partnerships would enable them to pool resources together to meet the requirements for bidding for grants and other proposal calls.

Third, the role of entrepreneurship is key in successfully implementing digital technologies and solutions for agricultural production. This also necessitates abandoning the focus on individual smallholder farmers and shifting attention to farmer cooperatives that can aggregate resources in securing the services of the enterprises. This is because individual farmers have been found to lack the capacity to secure such services due to high costs, high illiteracy rates and unwillingness to adopt digital technologies and solutions [58]. Therefore, it makes sense to not target small individual farmers but rather allow them to form communities or cooperatives, through which they can secure their collective power, meet the costs for digital services and draw the right government and development agency interventions. Another point of consideration is contagious farming, which includes land consolidation whereby individual smallholder farmers at the same location and farming the same crops consolidate their lands and secure the services of, for example, drones for spraying or scouting. In an economic sense, this would help both the farmers and the drone service provider. In this context, most of the interventions in Africa by governments and development agencies are undertaken by partnering with enterprises to carry out certain projects helping farming communities with their digital services. Some of the services include using drones to spray crops against the fall armyworm and other pests, profiling farmers and their needs and providing inputs, such as fertilizer, and related subsidies for these inputs.

Opportunities exist for old enterprises to explore and for new enterprises to tap into. That being said, there are still shortcomings that need to be addressed. The current state of drone regulations across Africa is not encouraging, as most enterprises still face various restrictions on operations. Internet and electricity coverage is limited when it comes to most rural communities, which impedes the smooth operation of digital services in those communities. These implications boil down to the reconfiguration of some of the solutions offered by the enterprises. For example, if a service or solution is provided through a web application, it will have to be reconfigured to feature a USSD code, allowing farmers to just dial a simple code to get access to the service instead of not having access to it because of a lack of internet connectivity.

### 5.1. Theoretical Contribution and Directions for Future Research

Our study adds to the existing literature on entrepreneurial ecosystems and communities of practices in digital agriculture [41,42]. It draws attention to the interest in entrepreneurship among youths in the digital agriculture space, which centres on a community of practice—AFGD—with a common goal of leveraging digital technologies and solutions to secure a brighter and prosperous future for agriculture in Africa. It also high-

lights the role of governments and states in supporting entrepreneurial ecosystems [59] and the learning and sharing mechanisms employed in such a community of practice in the attainment of the common goal [60,61].

Given these contributions, there are various opportunities for future research to explore. We highlight two of them here. The first future avenue for other scholars to look into relates to the collaborative mechanisms employed by enterprises in the formation of their business models in the digital agriculture space. Our case analysis showed that members of AFGD collaborate with each other and with developing agencies in executing certain projects. We did not, however, focus on how those collaborations influence the members' main business functions. This would, therefore, serve as an exciting area for future research. Another interesting avenue for future research relates to the transition from small-scale farming, which is centred on rural farmers, to large-scale farming, focusing on enhancement, inclusiveness and transparent agri-food systems that leverage digital innovations to meet food demands sustainably and secure better farming practices along with ecological sustainability. In this regard, determining what is needed for agricultural production to ensure a safe environment will further improve our understanding of the possible paths in the transition to sustainable agriculture.

### 5.2. Limitations

There are a few limitations to our study that we would like to discuss. The first concerns the depth of our topic modelling. We were limited to the information obtained about the services provided by the AFGD members. Other information, such as the types of business models and the company sizes, would have enriched our topic modelling and allowed us to detail the services provided and link them to the company size and its business model. The second limitation was the number of interviews we conducted. At the time we conducted the interviews, AFGD was evaluating and validating its members to confirm their continuity in the AFGD community. Therefore, only five members (stationed in five different countries—Ghana, Nigeria, Burkina Faso, Tanzania and Kenya) agreed to be interviewed. Furthermore, the period was a busy moment for most members, as they were fundraising and trying to secure projects. We believe the diversity in the regions of operations was represented in the outcomes of our case study.

### 6. Conclusions

The impact of digitalization in agriculture is huge and, in this study, we examined a community of practice (i.e., AFGD) involving several young enterprises that rely on digital technologies and solutions to provide cutting-edge services to farmers across Africa. We described the instances that led to the formation of this community of practice in which the role played by a development agency CTA was key. Being a community of practice, the enterprises relate to and collaborate as members of AFGD, and this has become a core aspect of inter-organization relationships fostering growth. Further, the members leverage social media platforms (WhatsApp and Twitter) to learn, share information and tell their success stories. As young enterprises, they are managed by young, educated individuals who are changing the status quo surrounding youth involvement in agriculture. The role of such a community of practice also extends to other actors in the ecosystem, who must collectively address the challenges in the agriculture sector and foster the development of an enabling environment for enterprises to successfully implement and deploy digital technologies for agricultural development in Africa. Although there have been improvements in the adoption of digital technologies for agriculture, (1) more education and capacity building or improvements are needed for farmers to expedite this adoption and (2) exploration of sustainable business models for digital solutions in the context of smallholder agriculture is needed to achieve various development targets within the respective African countries leveraging digitalization for agriculture.

**Author Contributions:** Conceptualization, M.A. and S.A.; methodology, M.A., A.L. and A.B.; validation, M.A., S.A., A.B. and A.L.; formal analysis, M.A., A.L. and S.A.; investigation, M.A.; resources, A.B. and A.L.; data curation, M.A. and A.B.; writing—original draft preparation, M.A.; writing—review and editing, M.A., A.L., S.A. and A.B.; supervision, S.A. and A.L.; project administration, M.A. and A.L. All authors have read and agreed to the published version of the manuscript.

**Funding:** This research received no external funding.

**Informed Consent Statement:** Informed consent was obtained from the key informants involved in this case study.

**Data Availability Statement:** Not applicable.

**Acknowledgments:** We would like to thank our key informants for their insights and for validating the findings of our desk analysis of AFGD.

**Conflicts of Interest:** M.A., S.A. and A.B. declare no conflict of interest. A.L. is the Chair of the Board at AFGD but his affiliation did not impact the analysis conducted in this case study.

## Appendix A

Interview guide

*Appendix A.1 Part 1: Information about the Interviewee*
1. Name and functions/position
2. Years worked in said position

*Appendix A.2 Part 2: Information about Africa Goes Digital (AFGD), Its Impact and Challenges*
3. What is AFGD?
4. When was AFGD found?
5. What circumstances led to the formation of AFGD?
6. How many enterprises are members of AFGD?
7. Which countries in Africa are the said enterprises from?
8. Which digital technologies are primarily used by AFGDs members?
9. Why is digitalization important for agricultural development in Africa?
10. What role are the members of AFGD playing in their respective countries in pushing the digitalization agenda?
11. What role is AFGD playing to push the digitalization agenda in Africa and to support its members sustain their businesses?
12. What role has been played by developing agencies and governments in driving the digitalization agenda for agricultural development in Africa?
13. What kind of collaborations exist with AFGD and developing agencies and governments across the globe aiming at the digitalization agenda for agricultural development?
14. i. What are some of the challenges for AFGD since its inception?
    ii. What are some of the opportunities for AFGD since its inception?

*Appendix A.3 Part 3: Information about AFGD's Members*
15. Do the members of AFGD meet to discuss, share ideas and experiences?
16. How often and by what means do they meet?
17. How are the members relating to each other in terms of knowledge and skills sharing?
18. What are the services provided by the members in their respective countries?
19. What are some of the challenges expressed by the members if there is any?
20. What has been done or what do you think can be done to address these challenges in order to enhance the services across Africa?
21. i. What are the challenges for new enterprises joining the digitalization agenda for agricultural development in Africa?
    ii. What are the opportunities for new enterprises joining the digitalization agenda for agricultural development in Africa?

## Notes

[1] Extant reports on digitalization in African agriculture can be found in the following links.

(a) The Digitalisation of African Agriculture Report 2018–2019. https://cgspace.cgiar.org/bitstream/handle/10568/103198/Executive%20 Summary%20V4.5%20ONLINE.pdf, URL (accessed on 28 December 2022).

(b) Toward digitalization futures in smallholder farming systems in Sub-Sahara Africa: A social practice proposal. Abdul-Rahim Abdulai *. https://www.frontiersin.org/articles/10.3389/fsufs.2022.866331/full. URL (accessed on 28 December 2022).

(c) Status of digital agriculture in 47 sub-Saharan African countries. https://www.fao.org/policy-support/tools-and-publications/ resources-details/en/c/1476537/. URL (accessed on 28 December 2022).

(d) Africa's Development regional Value Chains for a Sustainable Recovery Overview. https://read.oecd-ilibrary.org/view/?ref=11 27_1127899-pk0g8ydolw&title=Africa-s-Development-Dynamics-Overview&_ga=2.9854518.2052359467.1671911189-1594584702 .1671911189. URL (accessed on 28 December 2022).

[2] An overview of "Africa Goes Digital" can be found at their website http://www.afgoesdigital.com/. URL (accessed on 15 November 2022).

[3] The members of Africa Goes Digital have engaged in different sectors apart from agriculture, but in this study, we are focused on only the agriculture sector.

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
