# Peer review of "Developing Institutions and Inter-Organizational Synergies through Digitalization and Youth Engagement in African Agriculture: The Case of “Africa Goes Digital”"

_land, doi:10.3390/land12010199_

Round 1

Reviewer 1 Report (Previous Reviewer 2)

dear editor.

this version of the manuscript is much easier to understand, I only suggest a few small changes in :

1) the keywords, do not repeat the ones already in the title.

2) Adjust the titles of the tables and figures to the rules of the journal.

3) Include in the methodology a paragraph describing how and with what the graphs were made.

4) revise some subtitles of the RQ.

Greetings

Author Response

We thank you for your compliments and comments on our paper. We have incorporated your specific comments into the revised version of our paper.

  • We have removed keywords that appear in the paper’s title. Now, the keywords are “Digital innovations; Sustainable development; Digital agriculture; Self employment; Topic modelling; Entrepreneurial ecosystems; Community of practice; CTA”
  • You will also notice that we have adjusted the titles of tables and figures but because we used a template, it is a bit complicated when we try adjusting the tables and the figures and we believe this can be taken care of during production if the paper gets accepted.
  • In the data collection and analysis section of our methodology, we have included the following paragraph – “The outcome of our topic modelling is found in section 4.1 with the figures (Figure 1 to Figure 4) generated using Google Colaboratory (Google Colab). Bisong [55], examines the use of Google Colab for prototyping machine learning models”.
  • The subtitles of our RQs are now labelled as 4.1 RQ1...., 4.2 RQ2...... and 4.3 RQ3..... in the result section.

Kindly see the revised paper for your perusal. Thank you.  

Reviewer 2 Report (New Reviewer)

Your case study is a qualitative and interesting case study. This paper has an important conclusion for youth enterprises engaging in agriculture.

1.     The main question of this paper are:

RQ1: How was “Africa Goes Digital”(AFGD) established and the prevalent themes describing the services offered by its members leveraging digital innovations?

RQ2: How is AFGD promoting institutions and inter-organizational synergies for  its members to enhance their digitally-driven service to farmers?

RQ: What are the challenges and opportunities for both new and old enterprises in  the digital evolution of Africa’s agriculture?

2.The topic of this paper is original and  relevant, especially for  youth enterprises engaging in Africa’s agriculture.

The authors conclude that digitalization has increasingly importance in the agriculture sector in Africa because many African youths use digital technologies.

Also , the authors conclude that prior literature have paid less attention to the practice of youth enterprises engaging in agriculture through digitalization.

In this sense, they employ  ‘Africa Goes Digital’ (AFGD), a consortium of young African enterprises that provide services to farmers across the African continent.

Their results shows , for example, that:

(1) the Technical Centre for Agricultural and Rural Cooperation played an important role in supporting enterprises with drones ;

(2)  the lack of regulations for the use of drones is still a problem in the agriculture sector in Africa .

(3) It is important to discuss the implications of digitalization in supporting African governments in meeting sustainability goals.

(4) Though there are improvements in the adoption of digital technologies for agriculture, (1) more education and capacity building or improvement for farmers is needed ; (2) exploring sustainable business models for digital solutions in the context of smallholder agriculture is also needed.

Recommended  references can be:

The Digitalisation of African Agriculture Report https://www.cta.int/en/digitalisation-agriculture-africa

THE DIGITALISATION OF AFRICAN AGRICULTURE REPORT 2018–2019

https://cgspace.cgiar.org/bitstream/handle/10568/103198/Executive%20Summary%20V4.5%20ONLINE.pdf

Toward digitalization futures in smallholder farming systems in Sub-Sahara Africa: A social practice proposal

Abdul-Rahim Abdulai*

https://www.frontiersin.org/articles/10.3389/fsufs.2022.866331/full

Status of digital agriculture in 47 sub-Saharan African countries

https://www.fao.org/policy-support/tools-and-publications/resources-details/en/c/1476537/

AFRICA’S DEVELOPMENTREGIONAL VALUE CHAINS FOR A SUSTAINABLE RECOVERYOVERVIEW

https://read.oecd-ilibrary.org/view/?ref=1127_1127899-pk0g8ydolw&title=Africa-s-Development-Dynamics-Overview&_ga=2.9854518.2052359467.1671911189-1594584702.1671911189

Author Response

We are exceptionally grateful for your evaluation and compliments of our paper. We also appreciate the recommended references you gave. We have incorporated them into the revised paper more specifically as notes to support the following statement in the introduction – “Extant studies and institutional reports on digital technologies for agriculture forecast the potential of digitalization to drive the transition toward more sustainable agriculture [3-6]1”.

Reviewer 3 Report (New Reviewer)

It is an interesting paper that seeks to fill the gap in the literature. However, a revision is needed.  In particular, the methodology of the research paper could be enriched. More information on the selection of the sample is needed. By what criteria were the 11 enterprises and their representatives selected? Regarding the sample, its size is relatively small. Is it considered representative?

Author Response

We thank you for your evaluation and comments which have helped us to revise the paper.

We have provided more information on the selection of our key informants in section 3.2. However, based on your comments, we realized we have left out the limitations of our study in the previous version which we have now included in the revised version. We thank you for that. We however want to point a correction to your comment on “By what criteria were the 11 enterprises and their representatives selected”. As part of our findings, the 11 enterprises were those that got support from CTA and also founded AFGD, they were not our criteria for selection to take part in our study. Kindly refer to section 3.2 for more information about the data collection. Thank you.

This manuscript is a resubmission of an earlier submission. The following is a list of the peer review reports and author responses from that submission.

Round 1

Reviewer 1 Report

This writing introduced Africa digital ag development. The information will be good reference for the world to learn modern ag development. However, authors are suggested to address following concerns:

1) objectives of this study are not clearly presented

2) Methods: Did you collect the data or just used the data published by others such as [30]?

3) data need to be quantified or numeric. Only information is not meaningful to readers. For instance, what the percentage or how many farms are using UVA in which area?

4) Table 2: What that information will help readers and Africa farming?

Reviewer 2 Report

Dear

Editor, the article should be improved in all its sections so that it can be reconsidered for publication. The introduction section should be enriched, the methodology should be expanded to allow the replicability of the work. The results should include quantitative aspects and the results should be accompanied by graphs that allow a friendly and interesting reading of the document. 

I am sending an attachment with my general comments.

Regards
